# Factors influencing quality nutrition service provision at antenatal care contacts: Findings from a public health facility-based observational study in 21 districts of Bangladesh

Sk Masum Billah[1,2]*, Nazia Binte Ali[1], Abdullah Nurus Salam Khan[1], Camille Raynes-Greenow[2], Patrick John Kelly[2], Md. Shahjahan Siraj[1], Sufia Askari[3], Purnima Menon[4], Shams El Arifeen[1], Michael John Dibley[2], Phuong Hong Nguyen[4]

1 Maternal and Child Health Division, International Centre for Diarrhoeal Disease Research, Bangladesh (icddr,b), Dhaka, Bangladesh, 2 The University of Sydney School of Public Health, Sydney, New South Wales, Australia, 3 Children's Investment Fund Foundation, London, United Kingdom, 4 International Food Policy Research Institute (IFPRI), Washington, DC, United States of America

* billah@icddrb.org

**Data Availability Statement:** All relevant data are held on a public repository and can be accessed

## Abstract

Malnutrition during pregnancy is associated with increased maternal morbidity and mortality and has a long-term negative impact on child growth and development. Antenatal care (ANC) is the formal point of contact for pregnant women to receive preventive health and nutrition services. We assessed the quality of nutrition service delivery during ANC and examined its influencing factors related to the health facility, health care provider (HCP) and client characteristics. We conducted a cross-sectional assessment in 179 facilities, including 1,242 ANC observations and exit interviews of pregnant women from 21 districts in Bangladesh. We considered four essential nutrition services at each ANC contact including maternal weight measurement, anaemia assessment, nutrition counselling and iron-folic acid (IFA) supplement provision. We defined a composite 'quality nutrition service' outcome by counting the number of services (out of four) provided at each ANC from observation data. We explored both the supply-side and the client-level factors of quality nutrition service using multilevel Poisson regression. Overall, only 15% of clients received all four nutrition services. Performance of weight measurement (79%) was higher than IFA provision (56%), anaemia assessment (52%) and nutrition counselling (52%). The multivariable analysis showed that quality nutrition service delivery is positively associated with good logistical readiness of the facilities (aIRR: 1.23, 95% CI: 1.08–1.39), consultation by paramedics (aIRR 1.23, 95% CI: 1.06–1.42) and community health care providers (aIRR 1.32, 95% CI: 1.12–1.57), HCPs' knowledge on maternal nutrition (aIRR 1.04; 95% CI: 1.01–1.08), better HCP-client communication (aIRR 1.14; 95% CI: 1.04–1.26) and use visual aids or ANC card (aIRR 1.18; 95% CI: 1.11–1.27). We found limited associations between HCP training and external supervision with the quality of nutrition services. In conclusion, the quality of nutrition service provision during ANC is suboptimal. Public health nutrition programmers should

from https://doi.org/10.6084/m9.figshare.14608206.v1.

**Funding:** The lead author, SMB, received funding from Children's Investment Fund Foundation (CIFF) for conducting the initial assessment for the Evaluation of Accelerating Implementation of the National Nutritional Service (AINNS) project in Bangladesh (Grant number G-1612-00805). This paper uses the data from the initial assessment. Sufia Askari from the funding agency has contributed to the conception of the paper and provided inputs to the manuscript draft. The funding agency had no other role in original study design, data collection and analysis, decision to publish, or preparation of the manuscript.

**Competing interests:** I have read the journal's policy and the authors of this manuscript have the following competing interests: Sufia Askari is from the funding agency. She contributed to the study concept but played no role in study design, data collection or data analysis. Her contributions to this manuscript included inputs to the section on interpretation of results and review of manuscript drafts. However, the final decision about the results to include, interpretation and conclusion rested with the lead author and the authors from the evaluation team. All other authors declare that no competing interests exist. This does not alter our adherence to PLOS ONE policies on sharing data and materials.

ensure the facilities' logistical readiness, and revisit and reinforce the content and modality of training and supportive supervision of the HCPs. They should also emphasize positive HCP-client communication and the use of job aids to improve the quality of nutrition service provision during ANC.

## Introduction

Maternal nutrition is a major global public health concern. Undernutrition and micronutrient deficiencies during pregnancy are associated with increased maternal morbidity and mortality and have a long-term negative impact on child growth and development [1, 2]. Despite gradual progress in low and middle-income countries (LMIC) during the last two decades, maternal and child nutrition improvements remain an unfinished agenda, especially with the interruptions by the COVID-19 pandemic [1]. In Bangladesh, 42% of women of reproductive age (15–49 years) and nearly half of all pregnant women are anaemic, 12% are thin and 14% have short stature [3, 4]. The principal drivers for maternal malnutrition are poor-quality diets, low intake of micronutrient supplements and lack of access to quality healthcare during pregnancy [5, 6]. A comprehensive analysis of data from 69 LMICs suggested that receiving antenatal care (ANC) from appropriate sources and with recommended frequency was associated with positive birth outcomes and childhood nutrition [7]. Evidence also suggests receiving three or more ANC visits is likely to reduce maternal malnutrition by three-fold [8]. All these findings highlight the need for improving coverage and quality of nutrition services during pregnancy.

ANC is the first contact for pregnant women to health and nutrition services. "WHO antenatal care guideline for positive pregnancy experience" recommends routine supplementation of iron-folic acid (IFA), nutrition counselling and weight monitoring during ANC to ensure greater coverage of nutrition interventions during pregnancy [9]. Aligned with global recommendations, the Government of Bangladesh prioritized maternal nutrition service delivery using the ANC platform in their fourth Health Nutrition and Population Sector Program to continue 'mainstreaming nutrition' in the health systems [10]. Like several LMICs, timely utilisation and quality of ANC remain a major challenge to increase the reach and coverage of nutrition interventions in Bangladesh [11]. In 2017, the coverage of four ANC visits was 47%, indicating more than half of pregnant women were missing the opportunity of maternal nutrition services [4]. Another challenge is ensuring the quality of nutrition interventions provided at ANC contacts, without which programmes will not fully harness the benefits of these interventions on maternal and child nutrition outcomes. Moreover, a recent study reported incomplete readiness of healthcare facilities to provide nutrition interventions during ANC (51%) [12]. Nutrition input-adjusted coverage was suboptimal (18% for ANC) and disproportionately affected the poor and women with lower educational attainment. Suboptimal quality of care for nutrition services during ANC further deteriorates the situation. An assessment of the quality of nutrition services in primary care facilities showed that only 30% of mothers received four nutrition services during ANC [13].

The quality of health service delivery primarily depends on three levels of factors: structure (readiness of the health facilities), process (provision and experience of care) and outcomes (effective coverage and health outcomes) [14]. Previous studies from LMICs have mainly reported on the quality of ANC service by using one or some of the following indicators: frequency of visits, structural readiness to provide the service, content on maternity care, client experience and their determinants [11, 15–21]. Although most of these studies covered the

structural and outcome-level indicators, very few explored process-level indicators. One of the few attempts in exploring process-level factors and adherence to recommended quality provisions during ANC was made in 2016 in Tanzania [21]. The study reported that female providers, availability of routine tests and basic medicines at the public facilities and providers who received refresher training were positively associated with better adherence to quality ANC provision at a single point of contact. However, this study lacked the client-level factor and experience of care domains. Another study conducted in Ghana explored the client-level factors for quality ANC but lacked structural and provider-level factors [22]. A recent systematic review (2018) looking at the effectiveness of nutrition programming highlighted the need for additional analysis on barriers and enablers of nutrition service delivery using ANC and maternal, newborn and child health platforms in South Asia [23]. Considering the existing evidence and current research needs, a comprehensive analysis of quality ANC service delivery including structural, process and client-level factors, will be beneficial for effective programme planning [21]. This paper aims to assess the quality of nutrition service provision during ANC and examine its influencing factors related to facility readiness, health care providers, the process of care, and client characteristics.

## Methods

### Study design and setting

This study is a part of an initial cross-sectional assessment of coverage and quality of maternal and child nutrition services for a health system strengthening project, Accelerating Implementation of the National Nutritional Services (AINNS) in Bangladesh. The project included interventions to improve coordination of nutrition service delivery both at the national and sub-national levels and health care providers' (HCP) skills for maternal and child nutrition service delivery at public health care platforms. Details of the project evaluation plan have been reported elsewhere [24, 25]. Briefly, the initial assessment was done in 21 out of 64 districts in Bangladesh. We selected public health facilities from 14 of 40 AINNS intervention districts and seven of 24 non-intervention districts. We selected the districts based on comparability of population density, literacy rate, housing characteristics, people in the lowest wealth quintile, access to a safe drinking water source, electricity connection, improved sanitation, coverage of childhood immunization, ANC, skilled birth attendance, postnatal care and modern methods of family planning, under-five mortality rate and childhood stunting prevalence. We created a score for each district by principal component analysis of these variables. Then, we applied nearest neighbour matching of the district's PCA score to match two intervention districts with one non-intervention district. Finally, we selected seven matched groups of districts (each having two AINNS intervention districts and one non-intervention), which had the minimum difference in the PCA score.

The initial assessment included secondary to community-level public health facilities, namely district hospitals, sub-district hospitals, union health and family welfare centres and community clinics at the ward-level. At the district and sub-district-level hospitals, physicians with or without specialisation in obstetrics and gynaecology, nurses and paramedics provide maternal care services, including ANC. At union-level facilities, which serves a catchment of 25–30 thousand people, paramedics are the main ANC service providers. These paramedics have either a four-year medical diploma or 18 months of pre-service training including six months on midwifery skills. At community clinics, which serve six to eight thousand people in each ward, community health care providers and frontline community health workers provide ANC services. From each district, we selected the district hospital and two upazila health complexes (sub-district hospitals) from two randomly selected subdistricts for this assessment. We

also randomly selected four union-level health facilities and four community clinics from a list of functioning facilities prepared in consultation with the sub-district health manager in each chosen sub-district. We conducted the initial assessment between February and June 2016.

## Data collection

We used the structure and process domains of the Donabedian framework for the factors influencing quality nutrition service provision. Globally, the Donabedian framework is widely used for measuring the quality of healthcare service delivery [14, 26]. This framework identifies structure, process and outcome level indicators for a comprehensive measurement of quality of care [26, 27]. We have also considered the client characteristics as they have been found to influence the provision of ANC services [21]. We collected data at both supply and client-levels from different sources, including i) a health facility readiness assessment, ii) health care provider interviews, iii) direct observation of ANC practices, and iv) exit interviews with the ANC clients. For health facility assessment, we used a structured checklist adapted from the Bangladesh Health Facility Assessment tool [28] to collect information on facility infrastructure, and availability of specific logistics items (including weighing scale, haemoglobin testing tool, health education material/visual aids, ANC card, clinical guideline, registers and iron-folic acid (IFA) supplements). Interviews with ANC service providers included information about their duration of work in the facility, training status and knowledge regarding nutrition services. For direct observation, we developed the observation checklist following the standard operating procedures for maternal health services [29] to document the nutrition services provided during ANC consultations. Finally, we conducted exit interviews with clients after receiving ANC to collect information on their background, demographic and obstetric characteristics and ANC service care-seeking. S1 Table summarizes the outcome and explanatory measures, definitions and data collection methods.

## Sample size

We assessed the readiness to provide ANC services in 231 facilities (Fig 1). ANC services were not sought in 47 out of the 231 facilities on the assessment days; thus, we observed 1295 ANC service consultations at 184 facilities. We had complete background information from 201 HCPs who offered ANC services on the day of the visit. We excluded 53 observations from the analysis due to missing information on HCPs or the client characteristics. Finally, we included 1,242 observations of ANC service consultation provided by 201 service providers at 179 facilities in the analysis for this paper.

## Measures of quality of nutrition services (outcome measures)

We considered maternal weight measurement, screening for anaemia, nutrition counselling and provision of IFA supplements as the ANC nutrition services [10]. During the observations of ANC, the assessors recorded if these services were provided. We considered weight assessment performed if the HCPs took the weight of pregnant women using either a digital or analogue weighing scale. We included anaemia assessment in the quality of nutrition service during ANC due to the high prevalence of anaemia among women of reproductive age in Bangladesh [30]. Screening for anaemia included assessing blood haemoglobin levels either using Tallquist paper onsite, by previous laboratory investigation or examining eye or palm (clinical assessment). Provision of nutrition counselling included HCPs providing messages on dietary diversity, quantity and types of nutritious food. We defined IFA provision as HCPs distributing IFA supplements during the ANC consultation; however, we did not include HCPs prescribing IFA to be bought from outside pharmacies. We created a composite score variable for

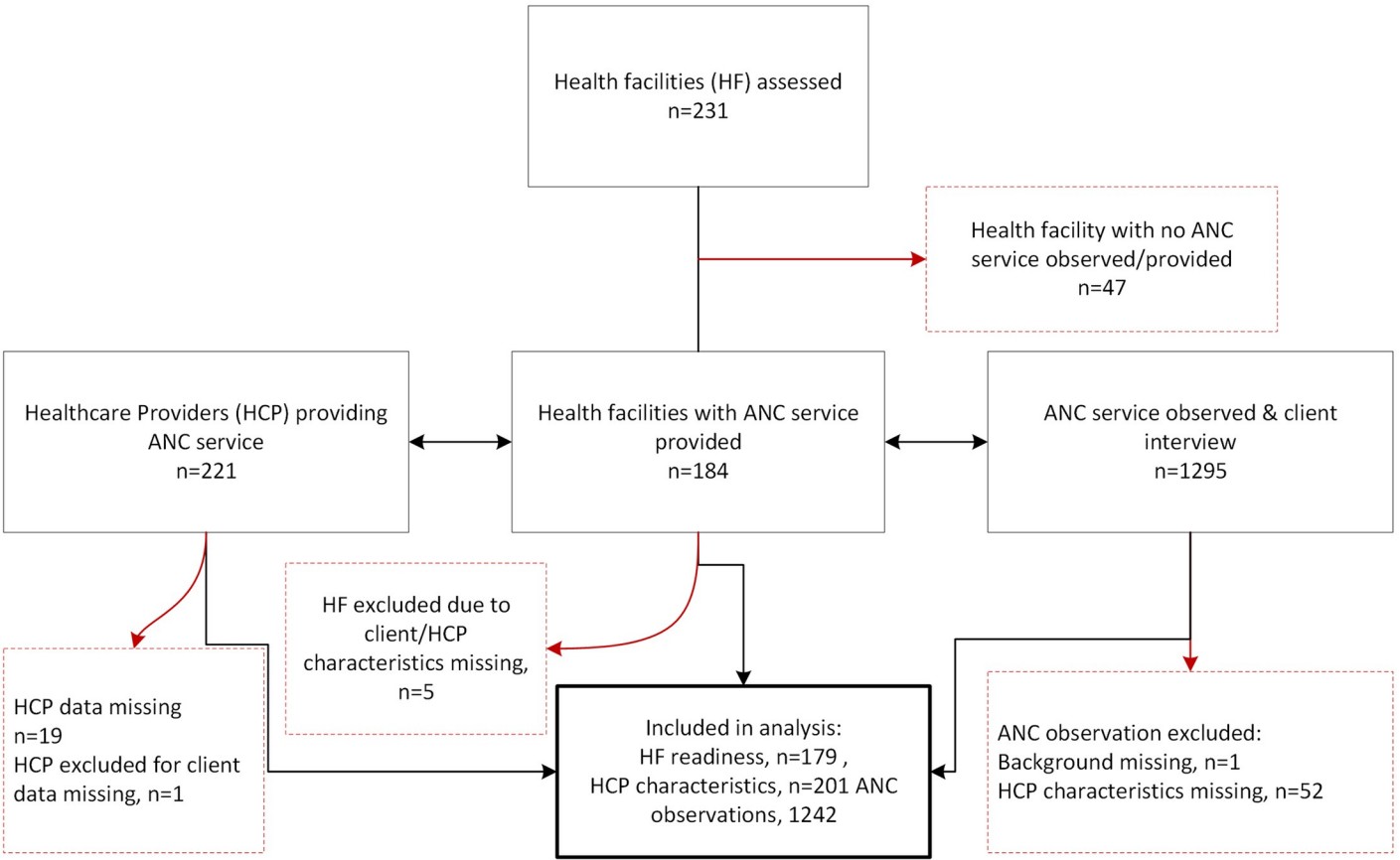

**Fig 1. Study flow diagram showing health facilities, health care providers and participants recruited and final sample size.** ANC: Antenatal care, HCP: Health care provider, HF: Health facility.

'quality nutrition service' by adding the number of services (out of four) provided at the ANC consultation. Similar methods of defining a composite score for quality service have been used in previous studies [11, 31].

## Explanatory variables

We explored the supply-side and the client-level factors of quality nutrition services provided at ANC (S1 Table). According to the 'structure' domain of the Donabedian framework [14], we considered the health facility characteristics and readiness and HCP characteristics and knowledge as supply-side factors. At the health facility-level, we constructed two indicators. The first was structural readiness (if all five readiness items were available). The second was the logistical readiness indicator created from seven logistic and supplies items using principal component analysis and then categorized into four logistical readiness quantiles. Cronbach's alpha for the logistics and supplies included in the composite score calculation was 0.62. HCP characteristics included age, sex, duration of service at the facility and receiving external supervision at least once in the last six months. HCPs' knowledge of nutrition services at ANC was a composite score derived by adding the correct responses to four questions (S1 Table). The process-level factor included two variables on HCP-client interaction: 'HCP-client communication' (HCP discussing the progress of pregnancy and asking if the client has any questions) and the 'use of visual aids and ANC card' during ANC service provision. Client-level factors

included client characteristics, namely age, education, gestational age and the number of ANC visits.

## Statistical analysis

We used descriptive analysis to summarise the client and provider characteristics, health facility readiness to provide nutrition services at ANC, and client-provider interactions during ANC consultation. We reported summary statistics by frequency and proportion or by mean (±SD) for categorical and continuous variables, respectively. We explored the differences in provider characteristics and provider-client interactions by types of providers and facility readiness by types of facilities using Analysis of Variance (ANOVA) and chi-square test for continuous and categorical characteristics variables, respectively. We checked for the linearity in the relationship between each continuous explanatory variable and nutrition service quality score by fitting the locally estimated scatterplot smoothing (loess) curve. As the outcome variable includes counts of nutrition services, we used Poisson regressions to examine the potential influencing factors of quality nutrition service at ANC [32]. We included provider and facility as the random effects in the model as clients were nested within the providers and providers were nested within the facilities. We calculated p-values using Wald tests with robust standard errors to adjust for model violations, such as under-dispersion. In the multivariable regression model, we included the explanatory variables that showed an association with quality nutrition service in the bivariate analysis at a p-value <0.2. The facility type had collinearity (correlation coefficient: 0.72) with HCP type. We compared Akaike's information criterion (AIC) and Bayesian information criterion (BIC) between the models with and without facility type and excluded facility type from the final model. We also ran multivariable Poisson regression using each of the four elements of quality nutrition services as the outcome. We reported the adjusted incidence rate ratios (aIRR) and their 95% confidence interval as the measure of association. Associations were considered statistically significant at a p-value<0.05 in the final model. We used Stata version 14 software in all statistical analyses.

## Ethical approval

We obtained ethical approval from the Ethical Review Committee of icddr,b (Protocol Number-15107), and written informed consent from HCPs, health facility managers and ANC clients before data collection.

## Results

The study includes data from 21 district hospitals, 40 sub-district hospitals, 61 union-level facilities and 57 community clinics. Most of the facilities (88%) had one HCP who provided ANC services on the day of assessment and participated in the study. Overall, 58% of facilities had all five infrastructural readiness items considered in this analysis (Table 1). Infrastructural readiness of community clinics (21%) and family welfare centres (56%) was poorer than higher-tier facilities, including district hospitals (95%) and sub-district hospitals (91%). The overall availability of supplies like weighing scales (82%), ANC registers (99%), IFA supplements (93%) and visual aids for counselling (88%) was high. However, only 35% of facilities had a haemoglobin testing kit to detect maternal anaemia and about one third had an ANC guideline. We found significant differences in the availability of haemoglobin testing kits, visual job aids, ANC cards and IFA supplements across different types of facilities (all p<0.01).

Most of the HCPs (86%) were female (Table 2). About half of the community HCPs received nutrition training compared to only 9% of physicians. Knowledge of nutrition services was similar across the HCP types. Although 84% of HCPs received external supervision

**Table 1. Characteristics and readiness of the healthcare facilities by types of facility.**

| | Overall (N = 179) | District hospital (N = 21) | Sub-district hospital (N = 40) | Union health centre (N = 61) | Community clinic (N = 57) | p-value |
|---|---|---|---|---|---|---|
| All structural readiness[1] items available | 57.5 | 90.5 | 95.0 | 55.7 | 21.0 | <0.001 |
| **Logistics and supplies** | | | | | | |
| Weighing scale | 82.1 | 81.0 | 72.5 | 85.3 | 86.0 | 0.318 |
| Haemoglobin testing tool | 35.2 | 9.5 | 17.5 | 50.8 | 40.4 | <0.001 |
| Health education materials/visual aids | 88.3 | 81.0 | 72.5 | 93.4 | 96.5 | 0.001 |
| ANC card | 84.9 | 71.4 | 72.5 | 95.1 | 87.7 | 0.004 |
| ANC guideline | 30.2 | 38.1 | 30.0 | 32.8 | 24.6 | 0.642 |
| ANC register | 99.4 | 95.2 | 100 | 100 | 100 | 0.056 |
| IFA supplement | 93.3 | 95.2 | 97.5 | 83.6 | 100 | 0.002 |
| **Average number of ANC services provided on a day** | | | | | | |
| <5 | 44.7 | 9.5 | 42.5 | 57.4 | 45.6 | <0.001 |
| 5–10 | 35.8 | 23.8 | 30.0 | 36.1 | 43.9 | |
| ≥11 | 19.6 | 66.7 | 27.5 | 6.6 | 10.5 | |

Values are %; p-values are test of difference in facility readiness across facility types;

[1] Structural readiness: Availability of examination room/area, privacy, examination bed, electricity and functioning toilet for clients

ANC: Antenatal care, IFA: Iron- folic acid

**Table 2. Characteristics of health care providers and provider-client interaction by types of health care provider.**

| | Overall (N = 201) | Physician (N = 35) | Nurse (N = 20) | Paramedics (N = 88) | CHCP/CHW (N = 58) | p-value |
|---|---|---|---|---|---|---|
| **HCP characteristics** | | | | | | |
| Age of provider (years) | 38.1 (±10.0) | 35.7 (±6.8) | 41.4 (±9.1) | 43.1 (±10.2) | 30.9 (±6.2) | <0.001 |
| Sex of the provider | | | | | | |
| Male | 13.9 | 5.7 | 0 | 4.6 | 37.9 | <0.001 |
| Female | 86.1 | 94.3 | 100 | 95.5 | 62.1 | |
| Duration of work at the current facility (years) | | | | | | |
| ≤3 | 32.3 | 62.9 | 35.0 | 35.2 | 8.6 | <0.001 |
| 4–7 | 37.8 | 25.7 | 15.0 | 15.9 | 86.2 | |
| ≥8 | 29.9 | 11.4 | 50.0 | 48.9 | 5.2 | |
| Received any in-service training on nutrition | 28.9 | 8.6 | 20.0 | 23.9 | 51.7 | <0.001 |
| Knowledge (score out of 6) on nutrition services at ANC[1] | 4.3 (±1.4) | 4.2 (±1.6) | 4.3 (±1.1) | 4.3 (±1.4) | 4.3 (±1.4) | 0.994 |
| Received any external supervision in last 6 months | 83.6 | 82.9 | 55.0 | 83.0 | 94.8 | 0.001 |
| Place of work | | | | | | |
| District hospital | 13.9 | 40.0 | 30.0 | 9.1 | 0 | <0.001 |
| Sub-district hospital | 26.9 | 60.0 | 70.0 | 21.6 | 0 | |
| Union health centre | 30.4 | 0 | 0 | 69.3 | 0 | |
| Community clinic | 28.9 | 0 | 0 | 0 | 100 | |
| **HCP-client interaction** | **N = 1242** | **N = 338** | **N = 179** | **N = 497** | **N = 228** | |
| HCP-client communication[2] | 72.6 | 61.8 | 63.1 | 87.7 | 63.2 | <0.001 |
| Used visual aids or ANC card | 47.0 | 29.9 | 71.5 | 56.3 | 32.9 | <0.001 |

Values are % or mean (± SD), p-values test difference of HCP characteristics by types of health care provider;

[1] includes knowledge on the provision of Iron-folic acid, diagnosis and management of anaemia, weight measurement and diet counselling;

[2] HCP discussing the progress of pregnancy, asked if the client had any question

HCP: health care providers, ANC: antenatal care, CHCP: community health care providers, CHW: community health workers

at least once in six months preceding the assessment, the external supervision of nurses was much lower (55%) compared to other types of HCP (>80%). HCP-client interaction during ANC consultation varied significantly by types of providers (Table 2). While the proportion of HCPs informing clients about the progress of their pregnancy or allowing clients to ask questions was high (73%), visual aid or ANC card usage during counselling was low (only in 47% of ANC consultations). Clients who received ANC from the facilities had a mean age of 23 years (SD±4.3 years) and had a mean eight years (SD±2.9 years) of schooling. Clients were mostly first-time pregnant women (42%), in their second trimester (46%), and came for either the second or third ANC visit (44%).

Among the four nutrition services to be provided during ANC, assessment of weight for maternal weight gain during pregnancy was the highest (79%), followed by IFA provision (56%) (Fig 2A). In nearly half of the ANC consultations, HCPs screened for anaemia either by testing haemoglobin or by clinical examination of eye or palm and provided counselling on appropriate dietary practices. Overall, only 15% of clients received all four nutrition services components and about one third received three services (Fig 2B).

We summarized the factors influencing the overall quality of ANC nutrition services in Table 3. The unadjusted results showed that logistical readiness of the facilities, types of HCP, knowledge of the HCP on maternal nutrition, provision for external supervision, HCP-client communication, use of visual aids or ANC cards during the consultation, and gestational age at ANC visit were positively associated with quality nutrition service at ANC. In the adjusted analysis, there was an association between facilities with average and good logistical readiness with higher quality nutrition service provision (aIRR: 1.17, 95% CI: 1.04–1.32 and aIRR: 1.23, 95% CI: 1.08–1.39, respectively) compared to facilities with low readiness. Although nurses provided services of similar quality to the physicians, paramedics were 23% (aIRR: 1.23, 95% CI: 1.06–1.42) and community health care providers were 32% (aIRR: 1.32, 95% CI: 1.12–1.57) more likely to provide quality nutrition services. Each point increase in HCPs' knowledge of nutrition was associated with a 4% higher quality of nutrition services at ANC (aIRR: 1.04; 95% CI: 1.01–1.08). There was no association between HCPs' training on nutrition and receiving external supervision in the last six months with quality nutrition service provision in the adjusted analysis. There was an association between HCP-client communication (aIRR: 1.14; 95% CI: 1.04–1.26) and the use of visual aids or ANC cards (aIRR: 1.18; 95% CI: 1.11–1.27) with quality nutrition service provision. Finally, the clients in the second and third trimesters of pregnancy received higher-quality nutrition service (aIRR: 1.23, 95% CI: 1.11–1.35 and aIRR: 1.19, 95% CI: 1.08–1.32, respectively) compared to those in the first trimester.

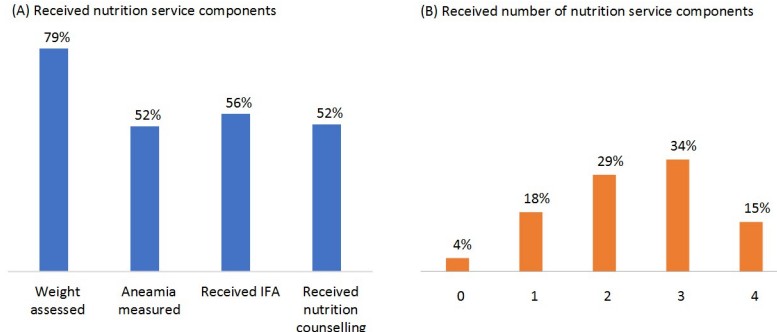

**Fig 2. Percentage of clients received (A) nutrition service components and (B) number of nutrition service components at ANC.** IFA: Iron-folic acid.

**Table 3. Unadjusted and adjusted incidence rate ratio of quality nutrition service at ANC according to health facility readiness, HCP characteristics, client-provider interactions and client characteristics.**

| Variables | Unadjusted models | | Adjusted model | |
|---|---|---|---|---|
| | IRR (95% CI) | p-value | aIRR (95% CI) | p-value |
| **Facility characteristics and readiness** | | | | |
| All structural readiness[1] available | 0.86 (0.77–0.96) | 0.007 | 1.00 (0.98–1.03) | 0.884 |
| Logistical readiness quantile[2] (ref: Level 1 (poor)) | | 0.015 | | 0.013 |
| Level 2 (average) | 1.23 (1.04–1.45) | 0.016 | 1.17 (1.04–1.32) | 0.011 |
| Level 3 (good) | 1.31 (1.11–1.54) | 0.001 | 1.23 (1.08–1.39) | 0.002 |
| Level 4 (best) | 1.27 (1.03–1.58) | 0.024 | 1.10 (0.95–1.28) | 0.201 |
| Average number of ANC services provided per day (ref: <5) | | 0.107 | | 0.302 |
| 5–10 | 0.92 (0.81–1.05) | 0.220 | 0.98 (0.89–1.07) | 0.614 |
| ≥11 | 0.85 (0.73–0.99) | 0.039 | 0.91 (0.82–1.02) | 0.123 |
| **Provider characteristics** | | | | |
| Age of provider (years) | 1.00 (0.99–1.00) | 0.447 | | |
| Sex of provider (ref: female) | 0.99 (0.84–1.17) | 0.919 | | |
| Duration of work at this facility (years) (ref: ≤3) | | 0.062 | | 0.731 |
| 4–7 | 1.16 (1.02–1.32) | 0.024 | 1.04 (0.93–1.16) | 0.537 |
| ≥8 | 1.13 (1.00–1.27) | 0.055 | 1.05 (0.93–1.18) | 0.464 |
| Type of provider (ref: physician) | | <0.001 | | <0.001 |
| Nurse | 0.99 (0.80–1.21) | 0.900 | 0.89 (0.74–1.05) | 0.169 |
| Paramedic | 1.41 (1.17–1.69) | <0.001 | 1.23 (1.06–1.42) | 0.006 |
| CHCP/CHW | 1.46 (1.21–1.76) | <0.001 | 1.32 (1.12–1.57) | 0.001 |
| Received any training on nutrition | 1.11 (1.00–1.24) | 0.060 | 1.00 (0.91–1.10) | 0.949 |
| Knowledge on nutrition service (score out of 6) | 1.07 (1.03–1.11) | 0.001 | 1.04 (1.01–1.08) | 0.012 |
| Received external supervision in last 6 months | 1.17 (1.01–1.36) | 0.034 | 1.09 (0.97–1.24) | 0.154 |
| **HCP-client interaction (process)** | | | | |
| HCP-client communication[3] | 1.24 (1.11–1.38) | <0.001 | 1.14 (1.04–1.26) | 0.007 |
| Used visual aids or ANC card | 1.19 (1.10–1.28) | <0.001 | 1.18 (1.11–1.27) | <0.001 |
| **Client characteristics** | | | | |
| Age of women (years) | 1.00 (0.99–1.00) | 0.169 | 1.00 (0.99–1.01) | 0.946 |
| Education (years of schooling) | 1.00 (0.99–1.01) | 0.781 | | |
| Gravida (ref: 1) | | 0.184 | | 0.188 |
| 2 | 1.01 (0.96–1.05) | 0.784 | 1.00 (0.96–1.05) | 0.841 |
| ≥3 | 0.95 (0.90–1.01) | 0.127 | 0.94 (0.87–1.02) | 0.149 |
| Gestational age (ref: 1st trimester) | | <0.001 | | <0.001 |
| 2nd trimester | 1.28 (1.16–1.42) | <0.001 | 1.23 (1.11–1.35) | <0.001 |
| 3rd trimester | 1.23 (1.10–1.37) | <0.001 | 1.19 (1.08–1.32) | 0.001 |
| Number of ANC visits (ref: 1st visit) | | 0.234 | | |
| 2nd–3rd | 1.05 (0.99–1.11) | 0.108 | | |
| ≥4th | 1.04 (0.99–1.09) | 0.156 | | |

[1] Structural readiness: availability of examination room, privacy, examination bed, electricity and functioning toilet for clients;

[2] Logistical readiness quantile was derived by categorizing score from principal component analysis of the availability of weighing scale, haemoglobin testing tool, health education materials/visual aids, ANC card, ANC guideline, ANC register and IFA supplement;

[3] HCP discussing the progress of pregnancy, asked if the client had any question;

IRR: incidence rate ratio, aIRR: adjusted incidence rate ratio, ANC: antenatal care, CHCP: community health care providers, CHW: community health workers; HCP: health care provider, ref: reference

**Table 4. Adjusted incidence rate ratio of weight measurement and anaemia assessment at ANC according to health facility readiness, HCP characteristics, client-provider interactions and client characteristics.**

| Variables | Weight measured | | Anaemia[1] assessed | |
|---|---|---|---|---|
| | aIRR (95% CI) | p-value | aIRR (95% CI) | p-value |
| **Facility characteristics** | | | | |
| All structural readiness[2] available | 1.00 (0.97–1.04) | 0.876 | 1.03 (0.96–1.10) | 0.483 |
| Logistical readiness quantile[3] (ref: Level 1 -poor) | | 0.002 | | 0.386 |
| Level 2 (average) | 1.61 (1.22–2.12) | 0.001 | 0.94 (0.67–1.33) | 0.749 |
| Level 3 (good) | 1.59 (1.21–2.11) | 0.001 | 1.18 (0.87–1.60) | 0.276 |
| Level 4 (best) | 1.39 (1.03–1.87) | 0.030 | 0.94 (0.63–1.40) | 0.761 |
| Average number of ANC services provided per day (ref: <5) | | 0.699 | | 0.163 |
| 5–10 | 1.00 (0.86–1.16) | 0.999 | 1.06 (0.81–1.38) | 0.668 |
| ≥11 | 1.06 (0.92–1.23) | 0.456 | 0.78 (0.57–1.07) | 0.130 |
| **Provider characteristics** | | | | |
| Duration of work in facility (years) (ref: ≤3) | | 0.486 | | 0.334 |
| 4–7 | 1.07 (0.90–1.28) | 0.429 | 1.17 (0.87–1.57) | 0.298 |
| ≥8 | 1.11 (0.94–1.30) | 0.212 | 0.95 (0.73–1.24) | 0.712 |
| Type of provider (ref: Physician) | | 0.001 | | 0.132 |
| Nurse | 0.94 (0.76–1.18) | 0.615 | 0.77 (0.51–1.17) | 0.222 |
| Paramedic | 1.23 (1.01–1.49) | 0.042 | 1.07 (0.80–1.43) | 0.658 |
| CHCP/CHW | 1.32 (0.99–1.74) | 0.055 | 0.76 (0.50–1.16) | 0.202 |
| Received training on nutrition | 0.95 (0.85–1.06) | 0.327 | 1.02 (0.76–1.37) | 0.882 |
| Knowledge on nutrition service (score out of 6) | 0.98 (0.93–1.03) | 0.404 | 1.18 (1.08–1.29) | <0.001 |
| Received external supervision in last 6 months | 1.00 (0.86–1.17) | 0.959 | 0.77 (0.60–0.99) | 0.042 |
| **HCP-client interaction (process)** | | | | |
| HCP-client communication[4] | 1.06 (0.92–1.20) | 0.427 | 0.87 (0.69–1.09) | 0.222 |
| Used visual aids or ANC card | 1.23 (1.11–1.37) | <0.001 | 1.28 (1.08–1.53) | 0.005 |
| **Client characteristics** | | | | |
| Age of women (years) | 1.01(1.00–1.01) | 0.144 | 1.00 (0.98–1.01) | 0.546 |
| Gravida (ref: 1) | | 0.084 | | 0.927 |
| 2 | 1.02 (0.96–1.08) | 0.546 | 0.99 (0.88–1.13) | 0.907 |
| ≥3 | 0.92 (0.82–1.02) | 0.120 | 1.03 (0.87–1.20) | 0.790 |
| Gestational age (ref: 1st trimester) | | 0.026 | | 0.128 |
| 2nd trimester | 1.19 (1.05–1.33) | 0.005 | 1.03 (0.85–1.25) | 0.763 |
| 3rd trimester | 1.19 (1.04–1.35) | 0.009 | 1.14 (0.93–1.38) | 0.206 |

[1] Anaemia assessment included HCPs assessing blood haemoglobin levels or examining eye or palm (clinical assessment);

[2] Structural readiness: Availability of examination room/area, privacy, examination bed, electricity and functioning toilet for clients;

[3] Logistical readiness quantile was derived by categorizing score from principal component analysis of the availability of weighing scale, haemoglobin testing tool, health education materials/visual aids, ANC card, ANC guideline, ANC register and IFA supplement;

[4] HCP discussing the progress of pregnancy, asked if the client had any question;

CHCP: community health care providers, CHW: community health workers; HCP: health care provider, aIRR: adjusted incidence rate ratio

We also explored facility, HCP and client-level factors influencing each of the components of nutrition services at ANC (Tables 4 and 5). The key influencing factors for weight measurement were the logistical readiness of the health facilities, clients seeing a lower-level provider like a paramedic or CHCP/CHW, using visual aids or ANC cards during service provision, and clients attending health facilities in the second and third trimester (Table 4). Anaemia assessment was also positively associated with higher HCPs' knowledge on nutrition and the

**Table 5. Adjusted incidence rate ratio of IFA provision and nutrition counselling at ANC according to health facility readiness, HCP characteristics, client-provider interactions and client characteristics.**

| Variables | IFA provided | | Nutrition counselling | |
|---|---|---|---|---|
| | aIRR (95% CI) | p-value | aIRR (95% CI) | p-value |
| **Facility characteristics** | | | | |
| All structural readiness[1] available | 0.97 (0.91–1.04) | 0.383 | 1.00 (0.94–1.06) | 0.996 |
| Logistical readiness quantile[2] (ref: Level 1 -poor) | | 0.410 | | 0.236 |
| Level 2 (average) | 0.83 (0.61–1.13) | 0.253 | 1.29 (0.96–1.75) | 0.095 |
| Level 3 (good) | 0.81 (0.62–1.05) | 0.108 | 1.35 (1.00–1.82) | 0.047 |
| Level 4 (best) | 0.84 (0.61–1.17) | 0.306 | 1.25 (0.90–1.74) | 0.190 |
| Average number of ANC services provided per day (ref: <5) | | 0.799 | | 0.045 |
| 5–10 | 0.93 (0.72–1.18) | 0.538 | 0.89 (0.72–1.10) | 0.271 |
| ≥11 | 1.01 (0.73–1.39) | 0.950 | 0.70 (0.53–0.94) | 0.016 |
| **Provider characteristics** | | | | |
| Duration of work in facility (years) (ref: ≤3) | | 0.454 | | 0.471 |
| 4–7 | 0.81 (0.58–1.14) | 0.235 | 1.15 (0.88–1.51) | 0.302 |
| ≥8 | 0.94 (0.68–1.33) | 0.779 | 1.18 (0.89–1.56) | 0.260 |
| Type of provider (ref: Physician) | | 0.023 | | 0.029 |
| Nurse | 0.87 (0.51–1.46) | 0.594 | 0.88 (0.57–1.38) | 0.586 |
| Paramedic | 1.41 (0.99–2.00) | 0.058 | 1.28 (0.89–1.85) | 0.182 |
| CHCP/CHW | 1.74 (1.19–2.56) | 0.005 | 1.56 (1.03–2.34) | 0.034 |
| Received training on nutrition | 1.11(0.88–1.40) | 0.357 | 1.01(0.85–1.20) | 0.886 |
| Knowledge on nutrition service (score out of 6) | 0.99 (0.91–1.07) | 0.745 | 1.08(1.01–1.16) | 0.040 |
| Received external supervision in last 6 months | 2.42(1.42–4.11) | 0.001 | 1.00(0.75–1.31) | 0.949 |
| **HCP-client interaction (process)** | | | | |
| HCP-client communication[3] | 1.34 (1.06–1.70) | 0.013 | 1.51 (1.17–1.95) | 0.002 |
| Used visual aids or ANC card | 1.05 (0.90–1.22) | 0.571 | 1.20 (1.01–1.42) | 0.038 |
| **Client characteristics** | | | | |
| Age of women (years) | 1.00(0.99–1.01) | 0.955 | 1.00(0.98–1.01) | 0.675 |
| Gravida (ref: 1) | | 0.239 | | 0.432 |
| 2 | 1.06 (0.96–1.15) | 0.245 | 0.96 (0.87–1.06) | 0.443 |
| ≥3 | 0.96 (0.84–1.11) | 0.599 | 0.90 (0.76–1.06) | 0.219 |
| Gestational Age (ref: 1st trimester) | | <0.001 | | 0.360 |
| 2nd trimester | 2.13 (1.58–2.88) | <0.001 | 0.92 (0.79–1.09) | 0.354 |
| 3rd trimester | 1.84 (1.35–2.51) | <0.001 | 0.89 (0.75–1.05) | 0.164 |

[1] Structural readiness: availability of examination room/area, privacy, examination bed, electricity and functioning toilet for clients;

[2] Logistical readiness quantile was derived by categorizing score from principal component analysis of the availability of weighing scale, haemoglobin testing tool, health education materials/visual aids, ANC card, ANC guideline, ANC register and IFA supplement;

[3] HCP discussing the progress of pregnancy, asked if the client had any question;

CHCP: community health care providers, CHW: community health workers; HCP: health care provider, aIRR: adjusted incidence rate ratio

use of visual aids or ANC cards but inversely associated with providers receiving external supervision. For IFA provision, the key influencing factors included lower-level HCPs like CHWs, clients in the second and third trimester of pregnancy, external supervision in the last six months, and HCP-client communication (Table 5). Provision of nutrition counselling was lower at facilities with 11 or more ANC services provided daily, but higher among community-level providers and HCPs with better knowledge of nutrition services. The use of visual aids or ANC cards and good HCP-client communication during the consultation also resulted in greater provision of nutrition counselling at ANC.

## Discussion

ANC is the formal point of contact for pregnant women to receive preventive health and nutrition services. We found that only 15% of women received all four nutrition services at ANC. Anaemia measurement and nutrition counselling were the least provided (52%) nutrition service. Logistical readiness of the facilities, types of ANC provider, HCPs' knowledge of nutrition services, client-provider interactions during the consultation and gestational age of clients were the factors influencing quality nutrition services during ANC. We found limited associations of training and external supervision of the providers with the quality of nutrition services. Our study adds to the knowledge about the quality of nutrition services provided during pregnancy. It provides some critical insights on areas to intervene for improving the nutrition service provision at ANC in a low-resource setting.

Our findings of the quality of nutrition service provision at ANC is consistent with the poor content of ANC services reported in Bangladesh and other low-income settings [11, 17, 20]. However, previous studies mostly focused on the content of ANC, asking women who had recently given birth if they had received any of the specified services at any point during their pregnancy, rather than on a specific service contact [11, 15, 20]. Studies from Peru and Tanzania observing-ANC service at the point-of-care reported low adherence to recommended service provision [19, 21]. A study in Bangladesh also reported high weight measurement and IFA provision but inadequate counselling on nutrition at ANC [13].

Adequate readiness of the health facilities is essential to ensure quality service provision [11, 33–35]. Our study showed that improved logistical readiness of health facilities is strongly associated with higher quality nutrition services provision at ANC. In the disaggregated analysis, we found weight assessment was substantially lower in the facilities with poor logistical readiness compared to the service at better equipped facilities. However, the provision of IFA did not differ significantly by the facilities' logistical readiness level as IFA was available in most (93%) of the facilities. Although unadjusted analysis showed an inverse association between infrastructural readiness of facilities and quality nutrition service provision, this association was attenuated and no longer significant in the adjusted analysis. The crude association was likely to be confounded by the higher-tier facilities with better infrastructural readiness but poorer service provision, as shown in our descriptive analysis. The National Nutrition Services programme should ensure a consistent supply of the logistics and supplies that are essential for nutrition service provision at ANC to improve the quality of service.

Lower-level providers who mainly offer services at lower-tier facilities provided a better quality of nutrition service than physicians and nurses at higher-tier facilities. Previous studies from LMICs including Bangladesh have also reported that lower-level providers had similar or better compliance with standard maternal and child out-patient care than higher-level providers [22, 36]. All physicians and nurses in our study provided services at the district and subdistrict hospitals where HCP shortage is a challenge [37]. These physicians and nurses often provide service under high workload pressure, are responsible for multiple tasks and manage patients with complications [38, 39]. In such situations, HCPs often prioritise curative health services over preventive nutrition services such as counselling and weight assessments for weight gain monitoring [39]. We also found nutrition counselling was lower at facilities where a higher number of ANC services were provided, and this occurs mostly at district and subdistrict hospitals. The role of the higher-level providers should be clarified based on WHO and national recommendations for nutrition services during pregnancy. Further in-depth qualitative assessment is necessary to better understand provider attitudes and motivation factors influencing their provision of nutrition services.

Our study demonstrates that HCP-client interactions such as informing them about the progress of pregnancy and giving them the opportunity to ask questions had a significant association with the quality of nutrition service provision. The WHO's standards for maternal and newborn care and guidelines on ANC recommend treating clients with dignity and compassion and providing information on health status and potential interventions for making an informed choice by clients [9, 40]. Earlier studies also reported the importance of HCPs' interpersonal communication and attitude towards clients to improve the quality of ANC services at health facilities [16, 18]. A study in Laos reported that HCP behaviour, mutual communication and treating clients with dignity are important considerations to improve ANC service quality [16]. This evidence highlights the importance of client-provider interactions for improving the quality of ANC nutrition services.

We also found that the application of job aids such as visual aids and ANC cards during ANC consultations improved the quality of nutrition services, especially for weight measurement, anaemia assessment, and nutrition counselling. A possible explanation for this is that documenting the key components of nutrition services provided on the ANC card, prompts HCP to provide the services [41]. Similarly, visual aids likely remind HCPs of the counselling contents, improved communication techniques and service provision [42]. However, there was no association between the use of ANC cards or visual aids and increased provision of IFA supplements. The ANC card used in public facilities in Bangladesh does not mention IFA provision. Moreover, IFA provision is more likely to depend on the availability of the supplements, HCPs perceptions about supplementation, and client characteristics.

In contrast to previous studies, we did not find any association between HCPs nutrition training and the quality of nutrition service delivery during ANC [21, 43]. The content and quality of their training in nutrition might explain this finding as there have been several criticisms of the government's Basic Nutrition Training that most HCPs received. There was too much content to cover in a short training time and it was overly focused on child feeding with few practical demonstrations [39]. The training quality declined when it was scaled up to a large number of HCPs nationwide using a cascade model [39, 44]. Only IFA provision was positively associated with receiving external supervision. Like training on nutrition, the quality of supervision visits to HCPs is often a question in resource-poor settings. HCPs often receive external supervision from administrative supervisors who have limited technical expertise and supervisory skills, resulting in a missed opportunity to reinforce the implementation of the knowledge and training obtained by HCPs into practice [39, 45]. Considering the quality issue of nutrition training, the current NNS operation plan introduced Comprehensive Competency-Based Training on Nutrition in 2017, which is implemented by technical experts [10]. The Competency-Based Training also addressed the supervision issue by developing and training supervisors on specifically designed modules to improve the technical competence of the supervisors. However, the success of Competency-Based Training in overcoming these challenges is yet to be evaluated.

Gestation was an important factor influencing the quality of nutrition services during ANC. Women coming for ANC in late trimesters received better quality nutrition services compared to women presenting in the first trimester [13, 22, 46]. The women presenting in the first trimester were less likely to have their weight assessed or receive IFA supplements. Low IFA supplement provision in the first trimester could be related to some inconsistent perceptions among HCPs on when to start IFA and the adverse effect of iron on foetal development [47, 48]. HCPs perception of the importance of different services and self-prioritisation of some services may also differ according to clients' gestational age. A study in Ghana suggested that HCPs sometimes disregard physical measurements in the first trimester [22]. However, the reasons for lower compliance with weight assessment in the earlier stage of

pregnancy require in-depth research. Detection and management of anaemia and provision of IFA early in pregnancy are important to avoid anaemia's detrimental effect on maternal and child health outcomes [49]. Adaptation of the national ANC guideline to comply with the global recommendations and improving HCPs' awareness of the recommended nutrition services during ANC visits may improve the quality of services.

The main strength of our study is the collection of observation-based data on nutrition service provision during ANC and the assessment of facility, provider and client characteristics at the same time-point, which allowed us to undertake supply-side and client-level factor analysis. The sampling of facilities covered 21 out of 64 districts spread across all geographic regions, which improved the generalisability of the study findings to public facilities in Bangladesh. Our study has some limitations. First, the observation of service provision by an external assessor may have resulted in some improved practice by HCPs. However, we anticipate that such bias will have a limited effect on the analysis of factors influencing the quality of service as facility and HCPs characteristics will not change during the observation. To reduce the Hawthorne effect, we conducted the observations in a natural setting, spending a reasonable time (i.e., one to two full days) at each facility [50]. We assured HCPs about their anonymity when presenting findings and emphasised that our observations were not related to individual performance evaluation. Second, the study methodology did not involve any in-depth interviews with HCPs to explore the reasons for the relatively low proportion of pregnant women whose weight was assessed during the first trimester of pregnancy and the limited impact of training and supervision on quality service provision. Third, we did not account for hospital management and governance level factors influencing the quality of nutrition service provision at health care platforms such as ANC. Fourth, we assessed only the on-site availability of rapid haemoglobin testing kits at the ANC room, which may underestimate the haemoglobin testing readiness of district and sub-district hospitals as some of them might have advanced diagnostic tests for anaemia. Future research should explore these questions.

## Conclusion

The quality of nutrition service provided at ANC in public health facilities in Bangladesh is sub-standard and varies by gestational age of the clients and type of HCPs proving the service. Increasing awareness of ANC standards and guidelines among HCPs is imperative. Logistical readiness of the facility influences the quality of nutrition service provision and demands urgent attention from nutrition programme managers. HCP-client communications and the use of simple but effective visual aids should be emphasised during ANC service. The limited impact of the traditional hierarchical administrative supervision calls for service-specific supportive supervision by technical experts to improve the quality of nutrition service provision at ANC.

## Supporting information

**S1 Table. Definition of outcome and explanatory variables and data collection methods.** (DOCX)

## Acknowledgments

We thank the Directorate General Health Services, Directorate General Family Planning and National Nutrition Services of the Ministry of Health and Family Welfare for approving us to conduct the study at public health facilities. We acknowledge the contribution of physicians and research assistants involved in the assessment. We are grateful to all participants for

allowing us to conduct the assessment and interviews. We would like to thank Julie Ghostlaw, a program manager at IFPRI, for her support to edit the manuscript. icddr,b is also grateful to the Governments of Bangladesh, Canada, Sweden and the UK for providing core/unrestricted support.

## Author Contributions

**Conceptualization:** Sk Masum Billah, Nazia Binte Ali, Abdullah Nurus Salam Khan, Camille Raynes-Greenow, Patrick John Kelly, Sufia Askari, Purnima Menon, Shams El Arifeen, Michael John Dibley.

**Data curation:** Md. Shahjahan Siraj.

**Formal analysis:** Sk Masum Billah, Nazia Binte Ali, Patrick John Kelly, Phuong Hong Nguyen.

**Funding acquisition:** Sk Masum Billah.

**Investigation:** Md. Shahjahan Siraj.

**Methodology:** Sk Masum Billah, Abdullah Nurus Salam Khan, Purnima Menon, Shams El Arifeen.

**Project administration:** Sk Masum Billah, Abdullah Nurus Salam Khan, Md. Shahjahan Siraj.

**Resources:** Shams El Arifeen.

**Supervision:** Sk Masum Billah, Abdullah Nurus Salam Khan, Camille Raynes-Greenow, Shams El Arifeen, Michael John Dibley.

**Visualization:** Sk Masum Billah, Md. Shahjahan Siraj, Phuong Hong Nguyen.

**Writing – original draft:** Sk Masum Billah.

**Writing – review & editing:** Nazia Binte Ali, Abdullah Nurus Salam Khan, Camille Raynes-Greenow, Patrick John Kelly, Md. Shahjahan Siraj, Sufia Askari, Purnima Menon, Shams El Arifeen, Michael John Dibley, Phuong Hong Nguyen.

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
