## [Decision Letter · Decision Letter 0]

28 Oct 2021

PONE-D-21-16534Determinants of quality nutrition service provision at antenatal care contacts: findings from a public health facility-based observational study in 21 districts of BangladeshPLOS ONE

Dear Dr. Sk Masum Billah,

Thank you for submitting your manuscript to PLOS ONE. After careful consideration, we feel that it has merit but does not fully meet PLOS ONE’s publication criteria as it currently stands. Therefore, we invite you to submit a revised version of the manuscript that addresses the points raised during the review process.

We look forward to receiving your revised manuscript.

Kind regards,

Sharon Mary Brownie

Academic Editor

PLOS ONE

Editor Comments 

Reviewers have offered a number of comments to enhance your script. Please carefully consider each comment and respond in full.

Journal Requirements:

3. Please note that PLOS does not permit references to 'data not shown.' Authors should provide the relevant data within the manuscript, the Supporting Information files, or in a public repository. If the data are not a core part of the research study being presented, we ask that authors remove any references to these data.

"I have read the journal's policy and the authors of this manuscript have the following competing interests: Sufia Askari is from the funding agency. She contributed to the study concept but played no role in study design, data collection or data analysis. Her contributions to this manuscript included inputs to the section on interpretation of results and review of manuscript drafts. However, the final decision about the results to include, interpretation and conclusion rested with the lead author and the authors from the evaluation team. All other authors declare that no competing interests exist." 

Reviewers' comments:

Reviewer's Responses to Questions

**Comments to the Author**

1. Is the manuscript technically sound, and do the data support the conclusions?

Reviewer #1: Yes

Reviewer #2: Yes

Reviewer #3: Yes

Reviewer #4: Yes

Reviewer #5: Yes

2. Has the statistical analysis been performed appropriately and rigorously? 

Reviewer #1: Yes

Reviewer #2: Yes

Reviewer #3: Yes

Reviewer #4: Yes

Reviewer #5: Yes

3. Have the authors made all data underlying the findings in their manuscript fully available?

Reviewer #1: Yes

Reviewer #2: Yes

Reviewer #3: Yes

Reviewer #4: Yes

Reviewer #5: Yes

4. Is the manuscript presented in an intelligible fashion and written in standard English?

Reviewer #1: Yes

Reviewer #2: Yes

Reviewer #3: Yes

Reviewer #4: Yes

Reviewer #5: Yes

5. Review Comments to the Author

Reviewer #1: The study has addressed an important public health issue for Bangladesh, and other LIMCs. However, one thing should be noted, the four factors which were finally revealed as independent determinants (logistical readiness of the facilities, HCP’s

knowledge on maternal nutrition, better HCP-client

communication, and use visual aids or ANC card) are essential pre-requisite for delivering quality nutrition service. If the pre-requisites are not fulfilled, it is not possible to deliver quality service. So, in discussion and conclusion sections of the papers should focused on the other factor (service provider: Doctors/paramedics/CHCP) as it is not a prerequisite like other four factors. Considering these issues, the authors should revised the term 'Determinants' used in the Title and introduction section.

Reviewer #2: ANC is the classic best entry point for counseling expecting mothers on proper nutrition, danger signs of pregnancy and monitor the health of the mother and the fetus. The authors looked at a critical health system fitness to provide quality nutrition services for pregnant women. The research questions are relevant and the methodology is sound. The results are clearly described and the implications well discussed. Just one curious question.

What did the authors think was the reason for the finding that the quality of nutrition services given during the 2nd and 3rd trimester was better than the one during the 1st trimester? I know that some misperception or misunderstanding from the provider side was mentioned. A bit more explanation would be great, if possible.

Reviewer #3: This manuscript will be a useful addition to our knowledge of nutrition services provided during pregnancy, especially for ANC in Low and Low Middle Income Countries.

I have a few clarification/suggestions that should be addressed;

Lines# 23 to 25. Definition is not clear. What is being counted? Will weight measurements done twice and anemia measurements done two times in one woman across pregnancy be counted as equivalent to weight measurement done once and and anemia measure done three times in another women?

Lines# 64 and 65. Some information is missing. Rephrase the statement for clarity.

Line#175. typo.

Table 1: Haemoglobin testing tool. Was it expected that sub-districts and community clinics will be better than district hospitals? Are they special programs/interventions that are being run by the health authority in Bangladesh that focus on resourcing rural and sub-district hospitals as compared to district hospitals? In most LMICS we see that district level hospitals are better equipped that rural clinics.

Table 1: What does "ANC utilization per day" mean?

What was the effect of clients (pregnant women) educational status on the outcome variables?

Figure 2 B: should the items add up to 100%? Currently it adds up 99%.

Reviewer #4: Methods

Upon what basis districts were selected?

Line 109:Better to mention Upa Zilla as sub district is non existent

Please mention how anaemia was defined

Conclusion:

Need to make brief depending upon objective

Reviewer #5: General comments

This manuscript is very interesting having very important findings, but the paper required same professional proofreading to edit the entire paper for language, grammatical, spelling, and punctuations to improve the paper for publication. Once the authors are able to address the concerns appropriately, the manuscript can be published.

The four main target variables (maternal weight measurement, anaemia assessment, nutrition counselling and iron-folic acid (IFA) supplementation) for this study was not properly addressed and exhausted, need to elaborate more.

Methods

Page 7, line 133 – 137 the paragraph “We conducted the assessment in 231 facilities, and in 184 facilities, we observed ANC services provided on the assessment days. We interviewed 217 health care providers who offered ANC services at the facilities on the day of the visit, and we observed 1296 ANC consultations. We excluded 54 observations due to missing information about the health care providers, the client’s characteristics, or the exit interview” is not clear, need to be reconsidered or rephrased.

Result

Page 10 line 213 & 2014 the sentence “On average, clients who received ANC from the facilities were 23 years old and had eight years of schooling” is not clear, need more explanation?.

Page 14, table 4 are highly congested and it crosses one page requires rearrangement or division into two table or more.

Discussion

Grammatical, spelling, and punctuations improvement is required in this section

Overall comment

There are some areas need corrections; such as rephrasing, grammatical, spelling, and language.

Kindly review issues raised and if possible get a trained English proof reader to edit the entire manuscript.

6. PLOS authors have the option to publish the peer review history of their article (what does this mean?). If published, this will include your full peer review and any attached files.

Reviewer #1: No

Reviewer #2: No

Reviewer #3: No

Reviewer #4: No

Reviewer #5: No

---

## [Author Response · Author response to Decision Letter 0]

27 Nov 2021

Response to reviewers' comments

Reviewer #1: 

1. The study has addressed an important public health issue for Bangladesh, and other LIMCs. However, one thing should be noted, the four factors which were finally revealed as independent determinants (logistical readiness of the facilities, HCP’s knowledge on maternal nutrition, better HCP-client communication, and use visual aids or ANC card) are essential pre-requisite for delivering quality nutrition service. If the pre-requisites are not fulfilled, it is not possible to deliver quality service. So, in discussion and conclusion sections of the papers should focused on the other factor (service provider: Doctors/paramedics/CHCP) as it is not a prerequisite like other four factors. Considering these issues, the authors should revised the term 'Determinants' used in the Title and introduction section.

Response: Our study hypothesizes that facility readiness, health care providers, the process of care, and client characteristics all contribute to the provision of high-quality nutrition services at antenatal care contact. We examined a variety of variables in each category (including provider types) and identified which of those variables are associated with (or may influence) high-quality service. We have revised our title and introduction, replacing the term “Determinants” by “Factors influencing”. In the discussion, we have included practice variations among different types of providers and essential programmatic inputs to mitigate the practice gap (page 19, lines 354-362) and now have included them in the conclusion (page 23 lines 444-445). 

Reviewer #2: 

1. ANC is the classic best entry point for counseling expecting mothers on proper nutrition, danger signs of pregnancy and monitor the health of the mother and the fetus. The authors looked at a critical health system fitness to provide quality nutrition services for pregnant women. The research questions are relevant and the methodology is sound. The results are clearly described and the implications well discussed. 

Response: Thank you for your comments on our manuscript. 

Just one curious question. What did the authors think was the reason for the finding that the quality of nutrition services given during the 2nd and 3rd trimester was better than the one during the 1st trimester? I know that some misperception or misunderstanding from the provider side was mentioned. A bit more explanation would be great, if possible.

Response: We have discussed some possible explanations in the discussion (page 21, lines 404-412). The difference in the quality of nutrition service across different trimesters of pregnancy is mainly due to the providers' perception gap and self-prioritisation of some services over others. For instance, among the four service components included in the quality of nutrition service during ANC, weight measurement and provision of IFA supplements were poorer in the 1st trimester compared to women in the 2nd and 3rd trimesters. Sometimes health care providers choose not to offer IFA in early pregnancy fearing iron’s adverse effect on foetal development. Similarly, sometimes they disregard physical measurement and weight assessment during early pregnancy. It is unlikely that the client’s demand or choice for service played a big role in this case, as clients often lack knowledge/perception of good quality care and what to anticipate from service providers when seeking health care in low and middle-income settings. 

Reviewer #3: 

This manuscript will be a useful addition to our knowledge of nutrition services provided during pregnancy, especially for ANC in Low and Low Middle Income Countries.

Response: Thank you for your encouraging comments on our manuscript. 

1. I have a few clarification/suggestions that should be addressed;

Lines# 23 to 25. Definition is not clear. What is being counted? Will weight measurements done twice and anemia measurements done two times in one woman across pregnancy be counted as equivalent to weight measurement done once and and anemia measure done three times in another women?

Response: We counted the number of the selected four services provided at each ANC consultation as the ‘quality nutrition service’ indicator from observation of the ANC consultations. It was a cross-sectional study, and we observed ANC service consultation for a woman only once. To avoid confusion, we have revised the text as:

 “We considered four essential nutrition services at each ANC contact including maternal weight measurement, anaemia assessment, nutrition counselling and iron-folic acid (IFA) supplement provision. We defined a composite ‘quality nutrition service’ outcome by counting the number of services (out of four) provided at each ANC from observation data” (page 2, lines 23-26). 

2. Lines# 64 and 65. Some information is missing. Rephrase the statement for clarity.

Response: We have rephrased the sentence to improve clarity (page 4, lines 65-68). 

“Another challenge is ensuring the quality of nutrition interventions provided at ANC contacts, without which programmes will not fully harness the benefits of these interventions on maternal and child nutrition outcomes. Moreover, a recent study reported incomplete readiness of healthcare facilities to provide nutrition interventions during ANC (51%).”

3. Line#175. typo.

Response: We have corrected the typo (page 10, line 203)

4. Table 1: Haemoglobin testing tool. Was it expected that sub-districts and community clinics will be better than district hospitals? Are they special programs/interventions that are being run by the health authority in Bangladesh that focus on resourcing rural and sub-district hospitals as compared to district hospitals? In most LMICS we see that district level hospitals are better equipped that rural clinics.

Response: In general, districts and sub-district level facilities have better readiness than union-level health facilities and community clinics. However, the availability of logistics and tests to provide ANC service is expected to be similar across all tiers of facilities. One explanation of poorer on-site readiness of haemoglobin testing at the district and sub-district hospital is out of the Tallquist strips due to high client load and availability of other advanced diagnostic facilities at district and sub-district levels. Health care providers sometimes refer the clients to these diagnostic labs for haemoglobin testing rather than doing the rapid test onsite. We did not consider the availability of advanced diagnostics tests for anaemia outside of ANC consultation as HCPs can recommend the test but do not take a decision if the test is done afterward. However, we have included the following text in the limitation (pages 22-23, lines 437-439)

 “Fourth, we assessed only the on-site availability of rapid haemoglobin testing kits at the ANC room which might under-estimate the haemoglobin testing capacities at district and sub-district hospitals as some of them may have advanced diagnostic tests for anaemia.” 

5. Table 1: What does "ANC utilization per day" mean?

Response: ANC utilization per day means the average number of ANC services provided on a day at the facility. We have now clarified this in Table 1 (page 11)

6. What was the effect of clients (pregnant women) educational status on the outcome variables?

Response: We did not find any association between the client’s educational status and the quality of nutrition service provided at ANC. We have reported this in table 3. Education was not associated with any of the four selected nutrition services in the bivariate analyses and we did not include it in the multiple regression model presented in Table 4. 

7. Figure 2 B: should the items add up to 100%? Currently it adds up 99%.

Response: We have corrected the rounding error in Figure 2B. 

Reviewer #4: 

1. Methods

Upon what basis districts were selected?

Response: We have expanded the text on district selection as follows (page 5-6, lines 103-111): 

“We selected the districts based on comparability of population density, literacy rate, housing characteristics, people in the lowest wealth quintile, access to a safe drinking water source, electricity connection, improved sanitation, coverage of childhood immunization, ANC, skilled birth attendance, postnatal care and modern methods of family planning, under-five mortality rate and childhood stunting prevalence. We created a score for each district by principal component analysis of these variables. Then, we applied nearest neighbour matching of district’s PCA score to match two intervention districts with one non-intervention district. Finally, we selected seven matched groups of districts (each having two AINNS intervention districts and one non-intervention), which had the minimum difference in the PCA score.” 

2. Line 109:Better to mention Upa Zilla as sub district is non existent

Response: We have added upazila health complexes and mentioned sub-district hospitals in parenthesis in line 123 on page 6. However, we have used sub-district hospital throughout the paper for the ease of understanding of the global audience. 

3. Please mention how anaemia was defined

Response: We have defined anaemia screening as (page 8, lines 169-171): 

“Screening for anaemia included assessing blood haemoglobin level either using Tallquist paper onsite, by previous laboratory investigation or examining eye or palm (clinical assessment)”. 

4. Conclusion:

Need to make brief depending upon objective

Response: We have revised the conclusion and made it relevant only to the study objectives and findings (page 23, lines 445-454). 

Reviewer #5: 

General comments

1. This manuscript is very interesting having very important findings, but the paper required same professional proofreading to edit the entire paper for language, grammatical, spelling, and punctuations to improve the paper for publication. Once the authors are able to address the concerns appropriately, the manuscript can be published.

Response: We have proof-read the manuscript and corrected typos and grammatical errors. 

2. The four main target variables (maternal weight measurement, anaemia assessment, nutrition counselling and iron-folic acid (IFA) supplementation) for this study was not properly addressed and exhausted, need to elaborate more.

Response: We have elaborated the definitions of four target variables for quality of nutrition service in lines 164-174 on page 8:

 “We considered weight assessment performed if the HCPs took the weight of pregnant women using either a digital or analogue weighing scale”. “Screening for anaemia included assessing blood haemoglobin level either using Tallquist paper onsite, by previous laboratory investigation or examining eye or palm (clinical assessment). Provision of nutrition counselling included HCPs providing messages on dietary diversity, quantity and types of nutritious food. We defined IFA provision as HCPs distributing IFA supplements during the ANC consultation; however, we did not include HCPs prescribing IFA to be bought from outside pharmacies”. 

3. Methods

Page 7, line 133 – 137 the paragraph “We conducted the assessment in 231 facilities, and in 184 facilities, we observed ANC services provided on the assessment days. We interviewed 217 health care providers who offered ANC services at the facilities on the day of the visit, and we observed 1296 ANC consultations. We excluded 54 observations due to missing information about the health care providers, the client’s characteristics, or the exit interview” is not clear, need to be reconsidered or rephrased.

Response: We have rephrased the text as (page 7-8, lines 149-157):

 “We assessed the readiness to provide ANC services in 231 facilities (Fig 1). ANC services were not sought in 47 out of the 231 facilities on the assessment days; thus, we observed 1295 ANC service consultations at 184 facilities. We had complete background information from 201 HCPs who offered ANC services on the day of the visit. We excluded 53 observations from the analysis due to missing information on HCPs or the client characteristics. Finally, we included 1,242 observations of ANC service consultation provided by 201 service providers at 179 facilities in the analysis for this paper”.

4. Result

Page 10 line 213 & 2014 the sentence “On average, clients who received ANC from the facilities were 23 years old and had eight years of schooling” is not clear, need more explanation?.

Response: We have rephrased the text as (page 12, lines 241-242):

 “Clients who received ANC from the facilities had a mean age of 23 years (SD±4.3 years) and had a mean eight years (SD±2.9 years) of schooling”.

5. Page 14, table 4 are highly congested and it crosses one page requires rearrangement or division into two table or more.

Response: We have divided Table 4 into two tables (Table 4 and Table 5). 

6. Discussion

Grammatical, spelling, and punctuations improvement is required in this section

Response: We have corrected grammatical, spelling and punctuation errors. 

7. Overall comment

There are some areas need corrections; such as rephrasing, grammatical, spelling, and language.

Kindly review issues raised and if possible get a trained English proof reader to edit the entire manuscript.

Response: We have revised accordingly.

---

## [Decision Letter · Decision Letter 1]

14 Dec 2021

PONE-D-21-16534R1Factors influencing quality nutrition service provision at antenatal care contacts: findings from a public health facility-based observational study in 21 districts of BangladeshPLOS ONE

Dear Dr. Sk Masum Billah,

Thank you for submitting your manuscript to PLOS ONE. After careful consideration, we feel that it has merit but does not fully meet PLOS ONE’s publication criteria as it currently stands. Therefore, we invite you to submit a revised version of the manuscript that addresses the points raised during the review process.

We look forward to receiving your revised manuscript.

Kind regards,

Sharon Mary Brownie

Academic Editor

PLOS ONE

Journal Requirements:

Reviewers' comments:

Reviewer's Responses to Questions

**Comments to the Author**

1. If the authors have adequately addressed your comments raised in a previous round of review and you feel that this manuscript is now acceptable for publication, you may indicate that here to bypass the “Comments to the Author” section, enter your conflict of interest statement in the “Confidential to Editor” section, and submit your "Accept" recommendation.

Reviewer #1: (No Response)

Reviewer #2: All comments have been addressed

Reviewer #3: All comments have been addressed

Reviewer #4: All comments have been addressed

2. Is the manuscript technically sound, and do the data support the conclusions?

Reviewer #1: Partly

Reviewer #2: Yes

Reviewer #3: Yes

Reviewer #4: Yes

3. Has the statistical analysis been performed appropriately and rigorously? 

Reviewer #1: Yes

Reviewer #2: Yes

Reviewer #3: Yes

Reviewer #4: Yes

4. Have the authors made all data underlying the findings in their manuscript fully available?

Reviewer #1: Yes

Reviewer #2: Yes

Reviewer #3: Yes

Reviewer #4: Yes

5. Is the manuscript presented in an intelligible fashion and written in standard English?

Reviewer #1: Yes

Reviewer #2: Yes

Reviewer #3: Yes

Reviewer #4: Yes

6. Review Comments to the Author

Reviewer #1: An interesting and important modifiable and independent service related factor revealed in your results is the TYPE OF PROVIDER. In your abstract, you have mentioned it nicely as- Although nurses provided services of similar quality to the physicians, paramedics were 23% (aIRR: 1.23, 95% CI: 1.06-1.42)

and community health care providers were 32% (aIRR: 1.32, 95% CI: 1.12-1.57) more likely to provide quality nutrition services. However, this factor has not been addressed at all in the discussion. As I have mentioned earlier (In the first review, this is the only important influencing factor/determinants and this need to be highlighted/explained in the DISCUSSION.

Reviewer #2: The study is useful. The findings are relevant. I re-confirm my opinion that the manuscript is clearly written, the methodology is sound and results are well discussed.

Reviewer #3: Go over the manuscript carefully to correct any grammatical issues and enhance ease of reading. For instance, you can summarize the conclusion to less than half the current length.

Reviewer #4: manuscript titled Factors influencing quality nutrition service provision at antenatal care contacts:

findings from a public health facility-based observational study in 21 districts of

Bangladesh. I had reviewed first and made few comments. All comments were addressed properly. It is satisfactory

7. PLOS authors have the option to publish the peer review history of their article (what does this mean?). If published, this will include your full peer review and any attached files.

Reviewer #1: No

Reviewer #2: No

Reviewer #3: No

Reviewer #4: No

---

## [Author Response · Author response to Decision Letter 1]

4 Jan 2022

Reviewer #1:

An interesting and important modifiable and independent service related factor revealed in your results is the TYPE OF PROVIDER. In your abstract, you have mentioned it nicely as- Although nurses provided services of similar quality to the physicians, paramedics were 23% (aIRR: 1.23, 95% CI: 1.06-1.42) and community health care providers were 32% (aIRR: 1.32, 95% CI: 1.12-1.57) more likely to provide quality nutrition services. However, this factor has not been addressed at all in the discussion. As I have mentioned earlier (In the first review, this is the only important influencing factor/determinants and this need to be highlighted/explained in the DISCUSSION.

Response: We agree with the reviewer that the type of HCPs is an important factor influencing the quality of nutrition service. We have now highlighted this in the discussion and proposed some possible explanations in the following additional text (page 19, lines 339-351):

 “Lower-level providers who mainly offer services at lower-tier facilities provided a better quality of nutrition service than physicians and nurses at higher-tier facilities. Previous studies from LMICs including Bangladesh have also reported that lower-level providers had similar or better compliance with standard maternal and child out-patient care than higher-level providers [1, 2]. All physicians and nurses in our study provided services at the district and subdistrict hospitals where HCP shortage is a challenge [3]. These physicians and nurses often provide service under high workload pressure, are responsible for multiple tasks and manage patients with complications [4, 5]. In such situations, HCP often prioritise curative health services over preventive nutrition services such as counselling and weight assessments for weight gain monitoring [5]. We also found nutrition counselling was lower at facilities where a higher number of ANC services were provided, and this occurs mostly at district and subdistrict hospitals. The role of the higher-level providers should be clarified based on WHO and national recommendations for nutrition services during pregnancy. Further in-depth qualitative assessment is necessary to better understand provider attitudes and motivation factors influencing their provision of nutrition services.” 

Reviewer #2: 

The study is useful. The findings are relevant. I re-confirm my opinion that the manuscript is clearly written, the methodology is sound and results are well discussed.

Response: We appreciate the positive comment from the reviewer. 

Reviewer #3: 

Go over the manuscript carefully to correct any grammatical issues and enhance ease of reading. For instance, you can summarize the conclusion to less than half the current length.

Response: We have checked the manuscript thoroughly and corrected grammatical errors and language. We have summarized the conclusion (Pages 22-23, lines 422-435). 

Reviewer #4: 

manuscript titled Factors influencing quality nutrition service provision at antenatal care contacts:

findings from a public health facility-based observational study in 21 districts of Bangladesh. I had reviewed first and made few comments. All comments were addressed properly. It is satisfactory. 

Response: We thank the reviewer for the encouraging comment. 

1. Amoakoh-Coleman M, Agyepong IA, Zuithoff NP, Kayode GA, Grobbee DE, Klipstein-Grobusch K, et al. Client factors affect provider adherence to clinical guidelines during first antenatal care. PLoS One. 2016;11(6):e0157542.

2. Hoque DE, Arifeen SE, Rahman M, Chowdhury EK, Haque TM, Begum K, et al. Improving and sustaining quality of child health care through IMCI training and supervision: experience from rural Bangladesh. Health Policy Plan. 2014;29(6):753-62.

3. Organization WH. Bangladesh health system review: Manila: WHO Regional Office for the Western Pacific; 2015.

4. Joarder T, Tune SNBK, Nuruzzaman M, Alam S, de Oliveira Cruz V, Zapata T. Assessment of staffing needs for physicians and nurses at Upazila health complexes in Bangladesh using WHO workload indicators of staffing need (WISN) method. BMJ Open. 2020;10(2):e035183.

5. Saha KK, Billah M, Menon P, El Arifeen S, Mbuya NV. Bangladesh National Nutrition Services: Assessment of Implementation Status: World Bank Publications; 2015.

---

## [Editor Report · Decision Letter 2]

7 Jan 2022

Factors influencing quality nutrition service provision at antenatal care contacts: findings from a public health facility-based observational study in 21 districts of Bangladesh

PONE-D-21-16534R2

Dear Dr. Sk Masum Billah,

We’re pleased to inform you that your manuscript has been judged scientifically suitable for publication and will be formally accepted for publication once it meets all outstanding technical requirements.

Kind regards,

Sharon Mary Brownie

Academic Editor

PLOS ONE

Additional Editor Comments 

Reviewer recommendations have been addressed.

---

## [Editor Report · Acceptance letter]

13 Jan 2022

PONE-D-21-16534R2 

Factors influencing quality nutrition service provision at antenatal care contacts: findings from a public health facility-based observational study in 21 districts of Bangladesh 

Dear Dr. Billah:

I'm pleased to inform you that your manuscript has been deemed suitable for publication in PLOS ONE. Congratulations! Your manuscript is now with our production department. 

Kind regards, 

on behalf of

Professor Sharon Mary Brownie 

Academic Editor

PLOS ONE